# Low Emulsifier Diet in Healthy Female Adults: A Feasibility Study of Nutrition Education and Counseling Intervention

**DOI:** 10.3390/healthcare11192644

**Published:** 2023-09-28

**Authors:** Mai A. Khatib, Haneen H. Saleemani, Nersian B. Kurdi, Haya N. Alhibshi, Manar A. Jastaniah, Sarah M. Ajabnoor

**Affiliations:** Clinical Nutrition Department, Faculty of Applied Medical Sciences, King Abdulaziz University, P.O. Box 80215, Jeddah 21589, Saudi Arabia

**Keywords:** food additives, emulsifiers, feasibility study, processed food, inflammatory bowel disease, Crohn’s disease

## Abstract

Emulsifiers are food additives commonly found in processed foods to improve texture stabilization and food preservation. Dietary emulsifier intake can potentially damage the gut mucosal lining resulting in chronic inflammation such as Crohn’s disease. This study investigates the feasibility of a low-emulsifier diet among healthy female adults, as no previous reports have studied the feasibility of such a diet on healthy participants. A quasi-experimental study for a nutrition education and counseling intervention was conducted over 14 days among healthy Saudi participants aged 18 years and over. Assessment of dietary intake using 3-day food records was conducted at the baseline and 2-week follow-up. Participants attended an online educational session using the Zoom application illustrating instructions for a low-emulsifier diet. Daily exposure to emulsifiers was evaluated and nutrient intake was measured. A total of 30 participants completed the study. At baseline, 38 emulsifiers were identified, with a mean ± SD exposure of 12.23 ± 10.07 emulsifiers consumed per day. A significant reduction in the mean frequency of dietary emulsifier intake was observed at the end of the intervention (12.23 ± 10.07 vs. 6.30 ± 7.59, *p* < 0.01). However, intake of macronutrients and micronutrients was significantly reduced (*p* < 0.05). Good adherence to the diet was achieved by 40% of the participants, and 16.66% attained a 50% reduction of emulsifier intake. The study demonstrates that a low-emulsifier diet provided via dietary advice is feasible to follow and tolerable by healthy participants. However, the diet still needs further investigation and assessment of it is nutritional intake and quality before implementing it in patients with inflammatory bowel disease who are at high risk of poor nutritional intake.

## 1. Introduction

Emulsifiers are a type of food additive widely used in processed foods [1]. They are used in food production to enable the formation and maintenance of a homogenous mixture with immiscible liquids, such as oil and water [2]. A huge selection of both synthetic and natural emulsifiers is available in the market. A manufacturer’s choice of emulsifier is largely dependent on the purpose of its use in food, whether for improving emulsion stability, thickening/texturing, increasing shelf life, or enhancing flavor [2]. Nearly 65 emulsifiers were identified in the literature for their common application in food [3,4]. As reported by the U.S. Food and Drug Administration, common examples of foods containing emulsifiers are chocolate, salad dressings, peanut butter, frozen desserts, and margarine [5]. The majority of foods consumed in the U.S. contain emulsifiers, which has been growing over time [6]. Overall, the role of food additives including emulsifiers is becoming increasingly significant with the rise in consumption of processed foods particularly in industrial countries with shifting lifestyles of the population [7].

When data concerning the consumption of emulsifiers were explored and compared against the prevalence rate of Crohn’s disease (CD), a questionable positive association was found [1]. Several possible mechanisms have been proposed to clarify the role of dietary emulsifiers in the pathogenesis of CD. Emerging evidence from animal and in vitro studies has shown that two widely consumed synthetic emulsifiers (polysorbate-80 and carboxymethylcellulose) have been linked with low-grade inflammation and the development of metabolic syndrome [8]. Alteration of the gut microbiome and the induction of colonic inflammation have also been reported as negative health impacts of high emulsifier intake [9]. Findings from several studies showed that emulsifiers can alter mucosal permeability and enhance bacterial translocation, which result in an active inflammatory response [10]. Increasing intestinal permeability leads to metabolic endotoxemia and low-grade systemic chronic inflammation, through the translocation of tryptophan and lipopolysaccharide-derived metabolites [11]. Studies on emulsifiers’ impact on the mechanisms discovered in in vivo and in vitro models have not yet been conducted on humans. Thus, the effect of dietary emulsifiers on health is not yet supported by data from human controlled trials.

Due to the possible involvement of food additives in illness etiology, studies exploring relationships between metabolic disorders and inflammatory bowel disease (IBD), and dietary additives as well as the therapeutic potential of anti-inflammatory [12,13] and additive-free diets in IBD [14,15], are becoming increasingly popular. A unique intervention in the form of a low-emulsifier diet was experimented recently by Sandall and colleagues in order to limit the consumption of all kinds of food emulsifiers [16]. In this study report, researchers evaluated quality of life related to food, nutritional intake, and symptoms of the disease in 20 patients with stable CD to determine if a low-emulsifier diet was feasible over the course of 14 days [16]. For the first time, a feasibility study showed that a low-emulsifier diet is tolerable and safe in CD patients, with encouraging results, including a 94.6% decrease in the frequency of eating foods that contain emulsifiers, a reduction in the symptoms related to CD, and an improvement in disease control scores. These results suggest that eliminating emulsifiers from one’s diet is feasible, despite the high degree of dietary behavioral change needed, such as changes in the planning of meals, preparation and shopping for food, and dining out, in order to adhere to such diet [16]. Although encouraging, this result might represent a placebo effect, which is impossible to be evaluated unless a control group is used [16]. Thus, more RCTs are required to verify the feasibility of such dietary interventions in healthy people and patients with IBD.

To date, no published reports have measured the feasibility and nutritional intake of a low-emulsifier diet and the average intake of foods containing food emulsifiers in healthy participants. Thus, the primary goal of this research was to examine the feasibility of following a low-emulsifier diet via a 2-week nutrition education and counseling intervention. This study would highlight the effects of such a diet before considering its implementation in patients with certain diseases, serve as a cornerstone for identifying the efficacy of the diet, and provide insight for future studies when considering a control group for randomized clinical trials [17].

## 2. Materials and Methods

### 2.1. Subjects

Healthy participants aged 18 years and over were recruited using a convenience sampling method. University students at King Abdulaziz University and their friends and relatives who were identified as healthy females were invited to enroll in the study. An online invitation was initially sent to the participants containing the study. Participants were identified as healthy based on their self-reported data entry. These included specific questions related to the exclusion criteria, which were identified by researchers and the literature to have been affecting the body response to food digestion and metabolism, and to ensure no negative effects to malnourished people or individuals with GI conditions. Exclusion criteria included individuals currently on special diets (i.e., low fermentable oligosaccharides or disaccharides, lactose-free, gluten-free, or vegetarian) and individuals with a food allergy or intolerance. Such individuals might have been already eliminating certain food items with high emulsifiers making them inappropriate for the feasibility study. Additionally, individuals with chronic and/or gastrointestinal diseases, pregnant or lactating females, and those with a body mass index below 18.5 kg/m^2^ were excluded because of reasons related to high nutritional risk.

### 2.2. Study Design

This was a non-controlled, quasi-experimental study for a nutrition education and counseling intervention that included assessments of dietary intake at a baseline and 2-week follow-up visit. The study took place between December 2021 and July 2022. The sample size was chosen in accordance with published guidelines, which recommended a minimum of 12 people [18,19], and was based on a previous feasibility study on IBD patient that used a similar intervention [16]. Thus, the target number was set at 30 participants to justify the study objectives and allow for any dropouts.

### 2.3. Baseline Demographic and Dietary Intake Data

After recording their demographic data, participants were asked to report all food and beverage intake with descriptions of amounts, brand names, and preparation methods. They were advised to select one day of the weekend and two weekdays when completing the 3-day food record. Written and verbal instructions were provided to improve the accuracy of the records. In addition, a research dietitian closely monitored the participants’ food records (via telephone/WhatsApp) and checked for missing data. Each completed food record was collected on a daily basis, to ensure accuracy of the information reported by participants. The Nutritics nutrition analysis software (Version 5.09, Dublin, Ireland) was used to assess dietary intake information from food records. This software has been used previously for its inclusiveness of several Middle Eastern food recipes and for its flexibility in manually inserting the ingredients when a specific food recipe is not found [20,21,22]. Four research dietitians were responsible for the dietary data collection and entry.

### 2.4. Nutrition Education and Counseling

After completing the baseline 3-day food diary, the participants were instructed virtually to limit their intake of foods containing dietary emulsifiers for 14 days, using the Zoom application. The dietary advice was based on excluding the 65 emulsifiers commonly used in food [3,4]. Dietetic counselling was conducted through an interactive online session, where research dietitians explained about emulsifiers, where they can be found, how to identify them in the ingredients list, and what substitutes are available. Educational material was provided to the participants, which helped them in identifying allowed food items and provided some recipes low in emulsifiers and practical shopping advice. Moreover, participants were able to communicate with three research dietitians for any inquiries regarding food choices. The participants were also advised to reduce the number of occasions of eating at restaurants during the study period.

### 2.5. Outcome Assessment

Frequency of exposure to dietary emulsifiers was assessed before and after commencing the diet using the 3-day food records. A comparison of baseline and end-of-intervention exposure to dietary emulsifiers served as a measure of adherence to the low-emulsifier diet. Participants were grouped into quartiles of adherence as follows: over 75% reduction, representing the high-adherence group; over 50% reduction, medium-adherence; over 10% reduction, poor-adherence; and 0% reduction or above, no-adherence. Nutrient intake was also assessed by comparing the estimated intake before and after the intervention. In addition, a brief diet-satisfaction survey was collected towards the end of the intervention. This was to assess the participants’ perceived acceptability of the diet after having restricted all classes of emulsifiers.

### 2.6. Dietary Emulsifier Intake Assessment

It was challenging to estimate the absolute intake of emulsifiers (i.e., mg/day) because there were no databases on food composition that showed the concentration of emulsifiers in foods. As a result, the participants’ daily exposure to emulsifiers was evaluated based on the consumption frequency, a method adopted from previous studies [16,23,24]. Every packaged food and drink item listed in the 3-day food diaries had its label examined by the research team for the presence of emulsifiers (qualitative data). All dietary emulsifiers from the 3-day food diaries were represented as a mean number of exposures per day (quantitative data). To determine the presence of emulsifiers in food products, we used the websites of national food and drink manufacturing companies and major Saudi Arabian grocery retailing companies (www.carrefourksa.com and www.luluhypermarket.com (accessed on 1 March 2022)), which contain food ingredients and composition data. For products suspected to contain emulsifiers with no available data, we used the CODEX General Standard for Food Additives (GSFA) online database, which describes food additives that have been evaluated for safety [25].

### 2.7. Ethical Approval

The Biomedical Ethics Research Committee at King Abdulaziz University in Jeddah, Saudi Arabia, approved the study (Approval number, HA-02-J-008). All the study participants gave their informed consent before participating.

### 2.8. Statistical Analysis

We used either the paired t test (for normally distributed data) or Wilcoxon signed rank test (for non-normally distributed data) to compare between the baseline and end of intervention for continuous variables. Adjusting for multiple comparisons was considered while performing multiple comparisons. GraphPad Prism (version 9.01, GraphPad Software, La Jolla, CA, USA) was used to prepare graphs. IBM SPSS Statistics for Windows, Version 28 (IBM Corp., Armonk, NY, USA) was used to perform the statistical analyses. A *p* value less than 0.05 was considered statistically significant.

## 3. Results

### 3.1. Characteristics of Study Participants

A total of 30 eligible healthy participants completed the low-emulsifier diet intervention and were included in the final analysis. No dropouts were reported. The participants’ mean age was 22.90 ± 5.84 years, and they were all females (Table 1).

### 3.2. Dietary Emulsifier Intake at Baseline

Study participants were found to be exposed to only 38 (of the 65) emulsifiers at baseline, with an average ± SD of 12.23 ± 10.07 emulsifiers consumed per day. The dietary emulsifiers with high-level exposure were diacetyltartaric and fatty acid esters of glycerol (0.93 ± 0.49), sodium phosphate (0.73 ± 0.89), polyphosphate (0.66 ± 0.92), and monoglycerides and diglycerides of fatty acids (0.64 ± 0.45) (Figure 1).

### 3.3. Main Food Groups Contributing to Emulsifier Intake at Baseline

Based on the CODEX, 29 food categories were identified at baseline from 213 foods containing emulsifiers consumed by the participants. The highest sources of dietary emulsifiers were bakery goods (34.7%) followed by dairy products (28.6%). However, confectionary products, beverages, and other products such as emulsified sauces and dips contributed less to emulsifier intake (8.9%, 8.5%, and 12.6%, respectively) (Table 2).

### 3.4. Effect of the Low-Emulsifier Diet on Dietary Emulsifier Intake

The mean frequency of total dietary emulsifier intake during the study decreased significantly from 12.23 ± 10.07 at baseline to 6.30 ± 7.59 at the end of the intervention (*p* < 0.01). A significant decrease in the mean frequency of exposure was observed for 16 emulsifiers (Table 3).

### 3.5. Effect of the Low-Emulsifier Diet on Nutrient Intake

At the end of the intervention, statistically significant reductions in the intake of nutrients were noticed, e.g., energy (mean difference of 447.52 ± 189.66 kcal, *p* < 0.05), carbohydrate (mean difference of 46.04 ± 10.68 g, *p* < 0.001), fat (mean difference of 19.10 ± 2.15 g, *p* < 0.001), and protein (mean difference of 21.14 ± 53.22 g, *p* < 0.001) (Table 4).

### 3.6. Adherence and Acceptability of the Low-Emulsifier Diet

Good adherence to the low-emulsifier diet was defined as a reduction of at least 75% in the intake of emulsifiers, and this was achieved by 40% (12/30) of the participants. Moreover, 16.66% (5/30) of the participants showed a 50% reduction in emulsifier intake frequency, and another 16.66% (5/30) showed 10% reduction. The remaining participants (30% (9/30)) showed an increase in exposure, signifying no adherence to the diet.

Among the reasons that contributed to adherence stated by more than 50% of the participants were that following the diet did not increase time spent preparing their meals, 56.66% (17/30); the flavor of meals and snacks was not less appetizing, 53.33% (16/30); and no extra expenses were spent on dining out and shopping for food, 53.33% (16/30). On the other hand, choosing appropriate foods when dining out was more challenging for over half of the participants (53.33% (16/30)), which may have contributed to less adherence (Table 5).

Data represent the frequency (n = 30) of 38 dietary emulsifiers exposure per day. CMC; Sodium carboxymethyl cellulose.

## 4. Discussion

The body of literature highlighting the negative impact of dietary emulsifiers on the development of metabolic and certain gastrointestinal diseases, such as CD, is still growing [1,8,9]. This has led to speculation that following a diet low in food emulsifiers can provide a means of protection against certain conditions and/or gastrointestinal diseases. A recent study with very promising results explored the effect of following a low-emulsifier diet on patients with CD [16]. The present research is the first to demonstrate the feasibility of implementing a low-emulsifier diet in healthy participants. In general, this study shows that delivering a low-emulsifier diet under dietetic supervision was effective in reducing exposure to dietary emulsifiers, as observed by the high acceptability and adherence rates.

At baseline, participants were exposed to 38 dietary emulsifiers every day. Following a diet that eliminates these food emulsifiers would require substantial lifestyle and dietary changes. However, the main contributing food groups were baked and dairy products. These results support earlier reports and emphasize the importance of bread as a dietary component among Saudi families [24,26,27]. Interestingly, in the present study, the total number of emulsifiers per day significantly decreased following the nutritional counseling intervention. Among them are monoglycerides and diglycerides of fatty acids, polysorbate 80, sucrose esters of fatty acids, and lecithin, which have been linked with the development of chronic diseases such as metabolic syndrome and IBD [1,8,23]. This is a promising result as it suggests that following a low-emulsifier diet can be effective in reducing risks in healthy participants, as Sandall et al. found in their study in relation to CD [16]. Furthermore, a good adherence rate was observed in the current study, making this diet feasible for implementation as both a preventive and a therapeutic intervention for healthy people and people with IBD in the future. This was further supported by the diet’s acceptability.

Nevertheless, extra care should be considered when trying to implement this diet as a lifelong diet in healthy participants or people with IBD, as in this study both macronutrient and micronutrient intakes were decreased following the intervention. A similar trend was observed in a previous report [16] as well as in other studies that utilized elimination diets as their main interventions [28,29]. This may indicate that by excluding commonly consumed foods containing dietary emulsifiers, participants may have unintentionally decreased their total intake, although an exchange list was provided with strict dietary advice to avoid change the quantity of their diet or try to lose weight during the period of the study. However, given that participants were also instructed to stay away from convenience foods and ready meals, this was expected logically, and could account for the drop in energy, salt, and saturated fat intake [30]. Since breads and rolls contributed the most to baseline emulsifier intake, compliance with the diet guidelines may account for the changes in nutrient intake. This may also explain the fall in carbohydrate consumption during the low-emulsifier diet. As suggested by Sandall et al., the diet significantly reduced emulsifier intake, but it also severely restricted other foods that did not contain emulsifiers, as emulsifiers are not always added to all grain products, which can unnecessarily result in nutritional deficiencies [16]. Moreover, it is important to keep in consideration the underreporting issue of energy and nutrients intake accompanying dietary food records [31] although it is considered as of the most accurate methods of dietary assessment [32]. Therefore, rigorous nutritional intake assessment is required, particularly for people with specific nutritional demands such as malnutrition and pre-existing dietary limitations.

Underestimation is another drawback to take into account in the current study given the absence of information on emulsifier quantity. Additionally, the total intake of emulsifiers from meals consumed in restaurants may not have been accurately assessed. For better evaluation of the nutrient and emulsifier content, specific restaurants could be evaluated for ingredient information, as Lee et al. suggested [23]. However, even the modest estimate used in the present study shows that emulsifier use is common and prevalent in this population. The same limitation was discussed in a previous study [16], and this should drive food manufacturers, companies, and restaurants towards better listing of food ingredients and food additives such as dietary emulsifiers in grams. It is a right of consumers to know exactly what kind of dietary constituents they are consuming, as this might change their dietary behavior and the quality of foods they consume [33]. It will also increase compliance with the diet.

A participant’s capacity to recognize the numerous emulsifiers included in foods while comprehending the large range of their existence in food items presents an additional challenge (for instance, some food companies incorporate emulsifiers in their breads; others do not). Delivering this nutritional and dietary information by certified dietitians through interactive sessions and handouts has enabled participants to identify the suitability of food products. Unlike using a digital application, as in Sandall et al.’s study [16], having human interaction throughout the course of the nutritional intervention allowed the dietitians to be in contact with the participants in everyday life in case any difficulties or concerns regarding the low-emulsifier diet arose.

It is worth noting that no dropouts were observed and all of the 30 participants completed the 14-day intervention. This further suggests the sufficient time participants had to point out any major issues associated with the diet with low burden on them and the acceptability of such a diet, as studies have shown that many people look for complementary and alternative therapies to treat their medical conditions, when present [34,35]. Indeed, the use of a food diary is superior to using dietary recall, as in the former participants record all foods and beverages in the diary to track oral intake, whereas techniques relying on memory and/or recall can be less effective [36]. This study largely highlights the feasibility and the applicability of a low-emulsifiers diet on healthy participants, who can serve as a control group to exclude any placebo response in future intervention studies [17].

## 5. Conclusions

In conclusion, the current study demonstrates that a low-emulsifier diet is tolerable in healthy participants and that emulsifiers are habitually consumed in everyday life. Although results exhibited a significant reduction in the intake of emulsifiers, this was accompanied by a reduction in the intake of micronutrients and macronutrients. Careful assessment of special nutritional needs is needed before implementing a low-emulsifier diet in the future. Future studies should consider implementing the diet on a representable sample of both males and females. Moreover, in order to assess the influence of implementing such a diet in various gastrointestinal disorders, recommendations to look into the impact of emulsifiers on gut inflammation at the cellular level and on the gut microbiota are needed.

## Figures and Tables

**Figure 1 healthcare-11-02644-f001:**
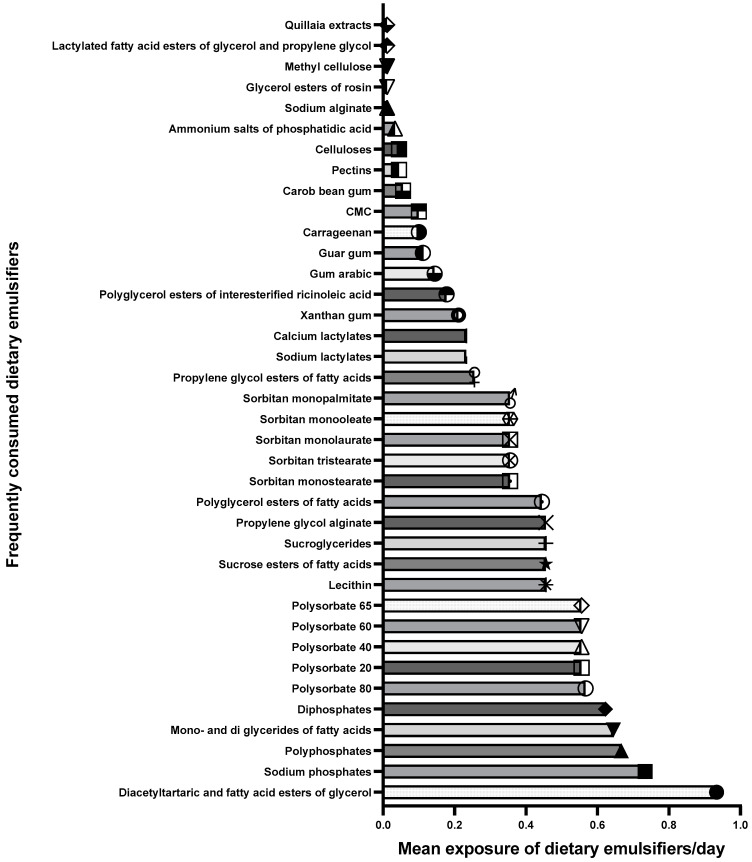
Baseline of mean exposure to 38 emulsifiers consumed per day by the study participants.

**Table 1 healthcare-11-02644-t001:** Baseline characteristics of participants.

Characteristics	n = 30
Sex, n (%): Female	30 (100%)
Age (year), mean ± SD	22.9 ± 5.84
Height (cm), mean ± SD	162.16 ± 6.75
Weight (kg), mean ± SD	61.53 ± 12.00
Region	
Riyadh	3 (10.00%)
Makkah	27 (90.00%)
Occupation	
Employed, n (%)	5 (16.66%)
Bachelor student, n (%)	19 (63.32%)
Housewife, n (%)	3 (10.00%)
Unemployed, n (%)	3 (10.00%)

SD = standard deviation. Data represent mean ± SD and/or n (%) of the study participants (n = 30).

**Table 2 healthcare-11-02644-t002:** Food groups contributing to frequency of dietary emulsifier exposure in habitual diet of 30 healthy participants at baseline.

Main Food Group	CODEX Food Category	CODEX Number	Contribution to Baseline Emulsifier Intake, n (%) *	Examples
Confectionery (8.9%)	Chocolate-based spreads	05.1.3	1 (0.5%)	Nutella
	Chocolate sweet products	05.1.4	15 (7%)	Twix
	Chewing gum	5.3	3 (1.4%)	Extra gum
Dairy products (28.6%)	Fluid milk	01.1.1	5 (2.3%)	Milk (long-life)
	Flavored fluid milk drinks	01.1.4	2 (0.9%)	Strawberry milkshake
	Condensed milk	01.3.1	2 (0.9%)	Condensed milk
	Milk powder	01.5.1	8 (3.8%)	Powdered milk
	Processed cream	01.4.2	3 (1.4%)	Whipping cream
	Processed cream/clotted cream	01.4.3	2 (0.9%)	Sour cream
	Dairy-based desserts	1.7	3 (1.4%)	Greek yogurt berries flavor
	Processed cheese	01.6.4	20 (9.4%)	Cheese slices and spreads
	Un-ripened cheese	01.6.1	14 (6.6%)	Mozzarella and halloumi
	Ripened cheese	01.6.2.1	1 (0.5%)	Cheddar
	Butter	02.2.1	1 (0.5%)	Butter
Beverages (8.5%)	Coffee, tea, and their substitutes	14.1.5	4 (1.9%)	2 in 1 coffee Nescafe
	Sport and carbonated drinks	14.1.4	9 (4.2%)	Spark drink and Pepsi
	Diet drinks or sugar substitutes	11.6	4 (1.9%)	Diet coke/Pepsi
	Fruit drink concentrate	14.1.4.3	1 (0.5%)	Sunquick
Processed meats (3.3%)	Processed meat	08.3.2	3 (1.4%)	Deli turkey meat slices
	Frozen processed meat	08.3.3	4 (1.9%)	Frozen burger patties
Processed vegetables and fruits (3.3%)	Vegetable and nut purees	04.2.2.5	4 (1.9%)	Peanut butter and tomato paste
	Processed fruits and products	04.1.2	3 (1.4%)	Jam
Bakery products (34.7%)	Breads and rolls	07.1.1	54 (25.4%)	Packaged supermarket bread
	Sweet bakery products	07.2.2	5 (2.3%)	Doughnut
	Cakes, cookies, and pies	07.2.1	15 (7%)	Digestive biscuits
Other products (12.6%)	Store-bought pasta or noodles	06.4.3	3 (1.4%)	Instant noodle
	Emulsified sauces and dips	12.6.1	15 (7%)	Mayonnaise
	Non-emulsified sauces	12.6.2	4 (1.9%)	Hot sauce
	Chips	15.1	5 (2.3%)	Lays chips

* Identified emulsifiers containing food at baseline (n = 213) were classified into food groups. Classification is based on the CODEX General Standard For Food Additives (GSFA) database, which describes food additives that have been evaluated for safety by the Joint Food Agricultural Organization/World Health Organization (FAO/WHO) Expert committee on Food Additives (JECFA) [25].

**Table 3 healthcare-11-02644-t003:** Mean frequency of dietary emulsifiers exposure pre- and post-intervention.

Emulsifiers	Baseline	Post-Intervention	*p*-Value
Diacetyltartaric and fatty acid esters of glycerol	0.93 ± 0.49	0.44 ± 0.43	0.00 ***
Sodium phosphate	0.73 ± 0.89	0.32 ± 0.49	0.00 **
Polyphosphates	0.66 ± 0.92	0.3 ± 0.50	0.01 **
Mono and di glycerides of fatty acids	0.64 ± 0.45	0.35 ± 0.38	0.00 **
Diphosphates	0.62 ± 0.86	0.32 ± 0.50	0.07
Polyoxyethylene 20 sorbitan monooleate (polysorbate 80)	0.56 ± 0.62	0.27 ± 0.36	0.03 *
Polyoxyethylene 20 sorbitan monolaurate (polysorbate 20)	0.55 ± 0.62	0.27 ± 0.36	0.03 *
Polyoxyethylene 20 sorbitan monopalmitate (polysorbate 40)	0.55 ± 0.62	0.27 ± 0.36	0.03 *
Polyoxyethylene 20 sorbitan monostearate (polysorbate 60)	0.55 ± 0.62	0.27 ± 0.36	0.03 *
Polyoxyethylene 20 sorbitan tristearate (polysorbate 65)	0.55 ± 0.62	0.27 ± 0.36	0.03 *
Sucrose esters of fatty acids	0.45 ± 0.53	0.21 ± 0.29	0.03 *
Sucroglycerides	0.45 ± 0.53	0.21 ± 0.29	0.03 *
Propylene glycol alginate	0.45 ± 0.52	0.24 ± 0.45	0.04 *
Lecithin	0.45 ± 0.38	0.15 ± 0.31	0.00 **
Polyglycerol esters of fatty acids	0.44 ± 0.54	0.21 ± 0.29	0.03 *
Sorbitan monostearate	0.35 ± 0.46	0.21 ± 0.29	0.15
Sorbitan tristearate	0.35 ± 0.46	0.21 ± 0.29	0.15
Sorbitan monolaurate	0.35 ± 0.46	0.21 ± 0.29	0.15
Sorbitan monooleate	0.35 ± 0.46	0.21 ± 0.29	0.15
Sorbitan monopalmitate	0.35 ± 0.46	0.21 ± 0.29	0.15
Propylene glycol esters of fatty acids	0.25 ± 0.41	0.20 ± 0.28	0.67
Sodium lactylates	0.23 ± 0.30	0.14 ± 0.27	0.25
Calcium lactylates	0.23 ± 0.30	0.14 ± 0.27	0.25
Xanthan gum	0.21 ± 0.39	0.13 ± 0.33	0.28
Polyglycerol esters of interesterified ricinoleic acid	0.17 ± 0.28	0.07 ± 0.22	0.11
Gum arabic	0.14 ± 0.28	0.01 ± 0.06	0.01 **
Guar gum	0.11 ± 0.23	0.04 ± 0.19	0.22
Carrageenan	0.10 ± 0.23	0.05 ± 0.21	0.47
Sodium carboxymethyl cellulose	0.10 ± 0.21	0.13 ± 0.28	0.78
Carob bean gum	0.05 ± 0.15	0.06 ± 0.20	0.86
Pectins	0.04 ± 0.11	0.01 ± 0.06	0.18
Celluloses	0.04 ± 0.11	0.02 ± 0.08	0.41
Ammonium salts of phosphatidic acid	0.03 ± 0.10	0.01 ± 0.06	0.31
Sodium alginate	0.01 ± 0.06	0.01 ± 0.06	1.00
Glycerol esters of rosin	0.01 ± 0.06	0.0 ± 0.0	0.31
Methyl cellulose	0.01 ± 0.06	0.01± 0.06	1.00
Lactylated fatty acid esters of glycerol and propylene glycol	0.01 ± 0.06	0.0 ± 0.0	0.31
Quillaia extracts	0.01 ± 0.06	0.0 ± 0.0	0.31
Total number of emulsifiers per day	12.23 ± 10.07	6.30 ± 7.59	0.00 **

Data represent mean frequency ± SD of dietary emulsifiers in diet of 30 participants. T-test was used for normally distributed data and Wilcoxon signed rank test was used for non-normally distributed data. *p*-value is considered significant if * ≤ 0.05; ** ≤ 0.01; *** ≤ 0.001.

**Table 4 healthcare-11-02644-t004:** Participants nutrients intake pre- and post-intervention.

Nutrient	Baseline	Post-Intervention	*p*-Value
Energy (kcal)	1576.08 ± 591.84	1128.56 ± 402.18	0.01 **
Protein (g)	76.22 ± 69.76	55.08 ± 16.54	0.00 ***
Fat (g)	62.88 ± 22.03	43.78 ± 19.88	0.00 ***
Carbohydrates (g)	167.46 ± 65.42	121.42 ± 54.74	0.00 ***
Sugars (g)	50.87 ± 23.49	36.62 ± 19.95	0.00 **
Fiber (g)	13.43 ± 5.26	11.92 ± 5.12	0.33
Saturated Fat (g)	20.35 ± 8.60	12.09 ± 5.86	0.00 ***
Monounsaturated Fat (g)	18.34 ± 7.87	15.82 ± 9.00	0.10
Polyunsaturated Fat (g)	10.13 ± 6.55	6.91 ± 3.64	0.00 **
Trans Fat (g)	0.43 ± 0.30	0.26 ± 0.25	0.02 *
Sodium (NA) (mg)	2141.20 ± 1236.76	1405.37 ± 1052.53	0.00 **
Potassium (K) (mg)	1802.52 ± 905.27	1465.54 ± 476.56	0.05
Calcium (Ca) (mg)	745.20 ± 728.47	544.52 ± 703.18	0.06
Magnesium (Mg) (mg)	176.58 ± 91.67	150.39 ± 46.40	0.11
Phosphorus (P) (mg)	841.61 ± 393.76	673.19 ± 206.01	0.03 *
Iron (Fe) (mg)	10.13 ± 4.45	9.11 ± 3.06	0.31
Copper (Cu) (mg)	0.71 ± 0.31	0.75 ± 0.57	0.95
Zinc (Zn) (mg)	6.03 ± 3.34	5.30 ± 1.95	0.39
Chloride (Cl) (mg)	1099.00 ± 1391.25	744.70 ± 1410.56	0.02 *
Manganese (Mn) (mg)	1.86 ± 0.78	1.64 ± 0.65	0.12
Selenium (Se) (μg)	79.26 ± 35.45	73.40 ± 30.17	0.44
Iodine (I) (μg)	8.47 ± 5.13	8.61 ± 4.63	0.24
Vitamin A (Total RE) (μg)	347.09 ± 473.88	372.97 ± 348.59	0.28
Vitamin D (μg)	1.89 ± 1.93	1.63 ± 1.44	0.22
Vitamin E (mg)	4.79 ± 3.74	4.99 ± 3.62	0.89
Thiamin (B1) (mg)	0.96 ± 0.45	0.99 ± 0.55	0.76
Riboflavin (B2) (mg)	1.10 ± 0.49	1.02 ± 0.48	0.44
Niacin (preformed) (mg)	17.90 ± 7.63	16.38 ± 6.22	0.28
Pantothenate (B5) (mg)	3.55 ± 1.60	3.23 ± 1.17	0.28
Vitamin B6 (Pyridoxine) (mg)	1.31 ± 0.74	1.29 ± 0.52	0.88
Biotin (B7) (μg)	5.99 ± 3.91	6.93 ± 2.77	0.59
Folate (B9) DFE (μg)	276 ± 164.43	283.20 ± 155.52	0.86
Vitamin B12 (Cobalamin) (μg)	2.56 ± 2.56	2.53 ± 3.41	0.55
Vitamin C (mg)	56.29 ± 46.10	49.35 ± 37.21	0.70

Data represent mean frequency ± SD of dietary emulsifiers in diet of 30 participants. T-test was used for normally distributed data and Wilcoxon signed rank test was used for non-normally distributed data. *p*-value is considered significant if * ≤ 0.05; ** ≤ 0.01; *** ≤ 0.001.

**Table 5 healthcare-11-02644-t005:** Participants’ acceptability of low emulsifier diet.

Response, n (%)	No	Slightly	Neutral	More	Much More
Meal preparation was more difficult	8 (26.66)	12 (40)	4 (13.33)	6 (20)	0 (0)
Longer time spent preparing and cooking meals	17 (56.66)	6 (20)	5 (16.66)	2 (6.66)	0 (0)
Longer time spent food shopping	11 (36.66)	10 (33.33)	2 (6.66)	7 (23.33)	0 (0)
Finding suitable foods when shopping was more difficult	10 (33.33)	7 (23.33)	5 (16.66)	9 (30)	1 (3.33)
Finding suitable foods when eating out was more difficult	2 (6.66)	12 (40)	1 (3.33)	16 (53.33)	4 (13.33)
The flavor of meals and snacks was less appetizing	16 (53.33)	10 (33.33)	3 (10)	1 (3.33)	0 (0)
More money spent on food shopping and eating out	16 (53.33)	9 (30)	5 (16.66)	0 (0)	0 (0)
The diet was more difficult	11 (36.66)	8 (26.66)	5 (16.66)	7 (23.33)	0 (0)
Following the diet for 6–8 weeks would be more difficult than normal	9 (30)	9 (30)	3 (10)	10 (33.33)	3 (10)

Data represent n (%) of responses of 30 participants.

## Data Availability

Data will be available upon reasonable request.

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
