# Peer review of "Low Emulsifier Diet in Healthy Female Adults: A Feasibility Study of Nutrition Education and Counseling Intervention"

_healthcare, 2023, doi:10.3390/healthcare11192644_

Round 1

Reviewer 1 Report

Dear Authors,

You wrote an interesting article demonstrating the feasibility of implementing a low emulsifier diet in healthy participants. The study found that providing a diet low in emulsifiers under the supervision of a dietitian was effective in reducing exposure to dietary emulsifiers, as observed by high rates of acceptance and adherence. The paper discusses a very important aspect of the increase in the consumption of processed foods, including emulsifiers and their impact on health.

The manuscript is well written and is well organized.

Some comments and suggestions:

1.     Small group size.

2.     In section 3.6. in the results section, the results are presented for a total of 31 individuals and should be 30 - because one person was discarded.

3.     Of concern is the large decrease in dietary calories after the 14-day intervention and the significant reduction in macronutrient and micronutrient intake noted by the authors, especially protein, calcium and iron. This begs the question: was the nutritional education sufficient? Did dietitians determine the recommended dietary macronutrient and micronutrient supply for healthy individuals before the intervention? It would be worthwhile to supplement this information? Future studies should certainly pay particular attention to this aspect, especially if such interventions lasted longer than 14 days.

4.     Include the digital object identifier (DOI) for all references where available.

Kind regards,

Author Response

Comment 1: - Small group size

-    This was a feasibility study in a new area of research, which serves as a proof of concept. Future research should consider the results of the current study when designing their protocol.

-    Comment 2: - In section 3.6. in the results section, the results are presented for a total of 31 individuals and should be 30 - because one person was discarded.

-    To clarify this better we have done all the analysis on the participants who met all the eligibility criteria for the feasibility study and who completed the intervention which are 30 participants. However, at the beginning of the result section (3.1), we mentioned that 31 participants were initially enrolled but one was excluded because of reason not meeting the eligibility criteria.

To avoid misunderstanding of the number of included participants, the results section (3.1) in the manuscript is now modified. 

-    Comment 3:- Of concern is the large decrease in dietary calories after the 14-day intervention and the significant reduction in macronutrient and micronutrient intake noted by the authors, especially protein, calcium and iron. This begs the question: was the nutritional education sufficient? Did dietitians determine the recommended dietary macronutrient and micronutrient supply for healthy individuals before the intervention? It would be worthwhile to supplement this information? Future studies should certainly pay particular attention to this aspect, especially if such interventions lasted longer than 14 days.

-    Thank you. This is an interesting point to mention. In the methods (Nutrition education and counseling section), the dietitians provided a practical dietary advice with a list of food substitutes. The dietitians have strongly advised participants not to change the habitual diet they were following – except with regards to emulsifiers intake. It is also important to point out that similar reductions in nutrients intake were encountered in elimination diet studies; including low emulsifiers diet study (Sandall et al, 2019). Meaning that it is hard to make sure that participants will restrict the intake of a certain food item without affecting their overall oral intake. Thus, indeed, future studies should address this matter and find a solution on how to control the intake of a regular diet while restricting certain food elements and without decreasing its quantity. The justification for this finding was provided in the discussion section (between Line 252 and 273)

-    Comment 4:- Include the digital object identifier (DOI) for all references where available.

-    Done. The updates reference list is attached to the reviewed manuscript with added DOI whenever available

Reviewer 2 Report

The manuscript investigated the feasibility of a low-emulsifier diet among healthy female adults. The introduction stated that it is the first article conducted among healthy participants. However, the introduction cites literature that is older. I think it is important to provide literature/data representative of the most current science on this topic. The authors should consider adjusting for multiple comparisons when conducting multiple tests to control the increased risk of Type 1 errors.

Abstract:

Line 9-11: may mention the gap in existing research

Line 17: specify the number of participants recruited initially and the number that completed the study.

Line 15: Mention the specific online platform used for the educational session (e.g., Zoom, Microsoft Teams).

Line 23-25: Expand on the implications of the study’s findings.

Introduction:

Consider providing literature/data representative of the most current science on this topic.

Line 63 -76: Looks like the whole paragraph is talking about Sandall and colleagues’ study. Consider reorganizing the paragraph to improve its flow and clarity (provide a logical progression of information from the broader context to the specific study and its implications)

Line 78-79: Provide a brief justification for why examining the feasibility in healthy participants is important. How might this information be relevant to broader health and nutrition discussions?

Materials and methods:

Line 90: specify how participants were identified as “healthy females” What criteria were used to determine their health status?

Line 91-95 The exclusion criteria are generally well-defined. However, you may want to briefly explain why these specific criteria were chosen. (e.g., adding the meaning of BMI below 18.5)

Line 97-99: It would be beneficial to briefly outline the key components of the nutrition education and counseling intervention.

Line 91: May use “single-arm study”?

Line 164: Consider adjusting for multiple comparisons when conducting multiple tests to control for the increased risk of Type 1 errors.

Discussion:

Line 267-268: It could benefit from specifying the potential consequences of this underestimation. How might it impact the study’s findings and their relevance?

Line 288-297: It would be beneficial to specify whether participants provided any feedback about their experience with the diet. Did they report any challenges or benefits?

Overall, the text demonstrates a good command of English, particularly in conveying scientific information. Minor refinements in sentence structure and word choice can further enhance its clarity and readability.

Author Response

-      Comment1 :- The manuscript investigated the feasibility of a low-emulsifier diet among healthy female adults. The introduction stated that it is the first article conducted among healthy participants. However, the introduction cites literature that is older. I think it is important to provide literature/data representative of the most current science on this topic. The authors should consider adjusting for multiple comparisons when conducting multiple tests to control the increased risk of Type 1 errors.
-    The reviewer has raised an interesting point. This was indeed the first kind of study to be conducted on healthy participants, however, previous report has investigated the effect of similar diet but on inflammatory bowel disease patients. With their absence of control group to base their conclusion on, it was important to fill this gap in the literature and test this diet on healthy participants. Thus, this study serves as a cornerstone for future research to inform researchers that it is feasible and applicable to apply this diet on healthy participants who can serve as control group in randomized clinical trials.
-    With regards to risks of type 1 errors, this was considered while performing the data analysis. This sentence will be added to the revised manuscript in lines 170-171: ‘’ adjusting for multiple comparisons was considered while performing multiple comparisons’’.

-    Comment 2:- Abstract: Line 9-11: may mention the gap in existing research
-    The following sentence will be added to line 12 in the modified manuscript: ‘’ , as no previous reports have studied the feasibility of such diet on healthy participants’’. 

-    Comment 3:- Abstract: Line 17: specify the number of participants recruited initially and the number that completed the study.
Thank you. It was decided that only the number of eligible participants who completed the study (n=30) will be mentioned in the results as well as in the abstract. The justification for this comment is provided in this letter in the previous comments by Reviewer 1 (Comment 2). 

-    Comment 4:- Abstract: Line 15: Mention the specific online platform used for the educational session (e.g., Zoom, Microsoft Teams).
-    Thank you for your comment. The following will be added to line 16 in abstract: ‘’using Zoom application’’ and will be mentioned in lines 126 to 128: ‘’After completing the baseline 3-day food diary, the participants were instructed virtually to limit their intake of foods containing dietary emulsifiers for 14 days, using Zoom application.’’

-    Comment 5:- Abstract: Line 23-25: Expand on the implications of the study’s findings.
Thank you for your comment. The research implications are now expanded in the modified abstract (Line 24-27)

-    Comment 6:- Introduction: Consider providing literature/data representative of the most current science on this topic
-    Thank you for your comment. I respect your opinion. Most of the supporting evidence for dietary emulsifier impact on health is based on preliminary research with limited data from human studies. Also, the introduction has more than half of its references that are published in the last 5 years. The limited number of trials in the field of dietary emulsifiers intake/restrictions makes it hard for the researchers to cite more than the references that have been already cited. The papers that have been generated from the time of write up to date are all review article that either support our conclusion, or out of scope of the manuscript objectives. Thus, it will not make significant edits in the introduction.

-    Comment 7:- Introduction: Line 63 -76: Looks like the whole paragraph is talking about Sandall and colleagues’ study. Consider reorganizing the paragraph to improve its flow and clarity (provide a logical progression of information from the broader context to the specific study and its implications)
-    The arrangement of sentences was changed accordingly to: ‘’ A unique intervention in the form of a low-emulsifier diet was experimented recently by Sandall and colleagues in order to limit the consumption of all kinds of food emulsifiers (16). In this study report, researchers evaluated quality of life related to food, nutritional intake, and symptoms of the disease in 20 patients with stable CD to determine if a low-emulsifier diet was feasible over the course of 14 days (16). For the first time, a feasibility study showed that a low-emulsifier diet is tolerable and safe in CD patients, with encouraging results, including a 94.6% decrease in the frequency of eating foods that contain emulsifiers, a reduction in the symptoms related to CD, and an improvement in dis-ease control scores. These results suggest that eliminating emulsifiers from one's diet is feasible, despite the high degree of dietary behavioral change needed, such as changes in the planning of meals, preparation and shopping for food, and dining out, in order to ad-here to such diet (16). Although encouraging, this result might represent a placebo effect, which is impossible to be evaluated unless a control group is used (16).’’

-    Comment 8:- Introduction: Line 78-79: Provide a brief justification for why examining the feasibility in healthy participants is important. How might this information be relevant to broader health and nutrition discussions?
-    This is one of the novel points about the current research, which was highlighted in lines 76 to 78, and in lines 83 to 86.

-    Comment 9:- Materials and methods: Line 90: specify how participants were identified as “healthy females” What criteria were used to determine their health status?

-    The following sentence will be added to lines 92-96 in the modified manuscript: ‘’Participants were identified healthy based on their self-reported data entry. These included specific questions related to the exclusion criteria, which were identified by researchers and the literature to have been affecting the body response to food digestion and metabolism, and to ensure no negative effects to malnourished people or individuals with GI conditions.’’ 

-    Comment 10:- Materials and methods: Line 91-95 The exclusion criteria are generally well-defined. However, you may want to briefly explain why these specific criteria were chosen. (e.g., adding the meaning of BMI below 18.5)

Thank you for this comment. This section in now modified in the method (Subject) section (Lines 92 to 96).

-    Comment 11:- Materials and methods: Line 97-99: It would be beneficial to briefly outline the key components of the nutrition education and counseling intervention.

-    All the components for the nutrition education and counseling intervention were outlined in details in lines 125 to 136.

-    Comment 12:- Materials and methods: Line 91: May use “single-arm study”?

-    True. However, the researchers think that this type of study is best defined under quasi experiment, where having a control group is possible but limited or was hard to find at the time of the study conduct

-    Comment 13:- Materials and methods: Line 164: Consider adjusting for multiple comparisons when conducting multiple tests to control for the increased risk of Type 1 errors.

-    With regards to risks of type 1 errors, this was considered while performing the data analysis. This sentence will be added to the revised manuscript in lines 170-171: ‘’ adjusting for multiple comparisons was considered while performing multiple comparisons’’

-    Comment 14:- Discussion: Line 267-268: It could benefit from specifying the potential consequences of this underestimation. How might it impact the study’s findings and their relevance?

-    The underestimation meant in this section was thought to be guided by the misreporting or underestimation of the emulsifiers content of foods either as in food labels, and/or in ready to eat services as in diners, where specific information on quantity and type of emulsifiers are not disclosed to the clients, leading to under-reporting and, thus, affecting study results.

-    Comment 15:- Discussion: Line 288-297: It would be beneficial to specify whether participants provided any feedback about their experience with the diet. Did they report any challenges or benefits?

-    The challenges reported were mentioned in lines 286 to 289.

-    Comment 16:- Overall, the text demonstrates a good command of English, particularly in conveying scientific information. Minor refinements in sentence structure and word choice can further enhance its clarity and readability.

-    Thank you for your comment, the manuscript was revised for any language related issues. 
